# Scaling Long Context Training Data by Long-Distance Referrals

**Yonghao Zhuang**[1] [*] **Lanxiang Hu**[2*]**, Longfei Yun**[2]**, Souvik Kundu**[3]**,**
**Zhengzhong Liu**[4]**, Eric P. Xing**[1,4]**, Hao Zhang**[2,5]
[1]Carnegie Mellon University, [2]University of California San Diego, [3] Intel, [4] MBZUAI,
[5] Snowflake

## Abstract

Training large language models for long context understanding faces the challenge of data shortage. Previous data engineering approaches mechanically concatenate short documents, which may create many pseudo long documents but raise concerns about data quality. In this paper, we study the core attribute of high quality data for long context training, and provide a data pipeline, LongPack, to scale such data. We found that *long distance referrals*, which occur in natural long documents, are crucial for long-context training. However, simply concatenating short documents does not reliably generate these relations. We further show that the density of long-distance referrals, which is higher in longer documents, has a key role in training efficiency, making previous upsampling methods suboptimal. To enrich long documents, we propose LongPack, a data pipeline that constructs long documents by packing shorter ones based on referral relationships. Specifically, for web pages, which are the primary source for language model training, we found hyper-link a native signal for such a relation. By packing web pages through their hyper-link connection, we can create longer, high-quality documents. Our experiments demonstrate that LongPack is highly scalable, generating a corpus of long documents equivalent in size to an entire pretraining dataset using just 0.5% root documents. Furthermore, the constructed documents have a 'near-natural' quality as innate long documents for long context training, reaching a 32.7% higher score than previous state-of-the-art methods.

## 1 Introduction

The demand for long-context understanding in large language models (LLMs) is steadily increasing. For instance, retrieval-augmented generation requires comprehension of passages with up to 32,000 tokens (Saad-Falcon et al., 2024); repository-level code completion typically involves contexts averaging 40,000 tokens (Liu et al., 2024b). However, attention mechanisms scale quadratically with sequence length, leading to significant computational overhead. This computational burden often makes pretraining on long contexts impractical. Instead, long-context understanding capabilities are usually acquired during continuous pre-training or post-training (Bai et al., 2023; Xiong et al., 2024), which we refer to as *long-context training*.

A major challenge in long-context training is the scarcity of high-quality data that matches the model's target context length, e.g., documents longer than 64K tokens make up less than 0.1% in a popular pretraining dataset( Figure 5a). As a practical remedy, short documents are concatenated, then truncated into chunks matching the desired context length (Xiong et al., 2024). Although this method mitigates data scarcity, but it raises concerns about the quality of these artificially concatenated chunks. Since tokens that are far apart often belong to different, irrelevant documents, models may struggle to learn true long-context dependencies from such data. Existing works (Li et al., 2023; Fu et al., 2024; Zeng et al., 2024) upsample long documents to mitigate the issue. However, due to the global shortage of very long documents, these methods still needs to use some "relatively long" documents to reach the demanded dataset size.

---

[*]Equal contribution. Part of the work was done when Yonghao and Lanxiang were interning at Snowflake. Contact: `yzhuang2@cs.cmu.edu`, `eric.xing@mbzuai.ac.ae`, `haozhang@ucsd.edu`

In this paper, we argue that simply upsampling relatively long documents is insufficient for optimal long-context training. We emphasize the importance of (super) *long distance referrals* – pairs of tokens that are semantically the same but separated by a significant distance within the document – for long-context training. We categorize referrals based on their distance and introduce the concept of long-distance referral density, which measures the number of such referrals normalized by the document length. Long documents tend to exhibit both higher density and greater distances in referrals, making them particularly valuable for long-context training. To validate this, we conduct experiments training the same model on sets of documents with similar quality but different distributions of referral distances. Models trained on documents with more and longer-distance referrals consistently perform better. Furthermore, disrupting these long-distance referrals results in a significant decline in performance.

Given the importance of long-distance referrals, using longer documents for long-context training is highly beneficial. However, the exponentially declining distribution of document lengths in existing datasets limits the availability of such long documents. To address this shortage, we propose Long-Pack, a *scalable* data pipeline that constructs *high quality, sufficiently long* documents. Our approach leverages the fact that many pretraining datasets, such as C4 (Dodge et al., 2021) and Dolma (Soldaini et al., 2024), are derived from web pages, which often contain numerous hyperlinks. These hyperlinks serve as natural indicators of strong referral relationships between web pages. For each web page, we extract its hyperlinks from the raw HTML and retrieve the linked content. We then clean and concatenate the retrieved content to form long documents. As these long documents are structurally built based on referral relationships, they preserve or even enhance the density of long-distance referrals as found in naturally long documents. Long-context training with these packed data improves 32.7% over previous data generation recipe.

In summary, we introduce a novel data construction pipeline to address the scarcity of long documents for long-context training. We demonstrate that long-distance referrals are a key attribute of high-quality long documents. Based on the importance of long-distance referrals, we accordingly propose LongPack to solve the shortage of long enough documents. Our approach provides a scalable solution for constructing long documents. This ensures a high standard of data quality that is essential for long-context training. Experiment result shows that, with only 0.5% data, we can generate a dataset with long documents as many as the whole pre-train dataset. On 5 million documents randomly sampled from the pre-train dataset, we make 11.3% of them grows 13 times larger in average. Moreover, compared to the previous state-of-the-art upsampling long document method (55.1), our method reaches a score of 73.0 in the comprehensive evaluation for long-context understanding.

## 2 BACKGROUND AND RELATED WORK

**Algorithm**  Language models trained with shorter maximum context lengths, typically 4096 or 8192 tokens, cannot easily adapt to tasks requiring longer contexts (Xiong et al., 2024). The ideal solution would be to pretrain the model with a larger context length. However, attention mechanism has a quadratic complexity relative to sequence length, making such pretraining computationally expensive. A two-stage training process is therefore employed. First, the model is pretrained with a short context length, typically 4096 or 8192, using trillions of tokens; After that, it is trained to adapt to a much larger sequence length, like 128K or 1M, with fewer tokens. Recent studies (Chen et al., 2023; kaiokendev, 2023) found that during the second stage, position embedding plays an important role. Rather than extrapolating embeddings, interpolation proves more effective. Interpolation means the new position embedding factor at $i$ equals that at $i/k$ under the original context length, where $k$ represents the ratio of the new maximum context length to the original one.

**Data corpus**  Several studies have investigated the data requirements for developing models with long-context understanding. Intuitively, long-context training requires long sequence data, leading many works to focus on domains rich in such content: LongChat (Li et al., 2023) is trained on long dialogues, and YaRN (Peng et al., 2023) on books. However, these data is insufficient in both length and quantity. For example, The Pile (Gao et al., 2020) only has 6.3GB book data. Besides, Fu et al. (2024) found that while these datasets improve long-context understanding, domain shifts during continued training can significantly degrade the model's overall performance.

On the other hand, directly sampling long documents from the pretraining dataset faces the data shortage issues. For example, in Figure 5a, we analyzed RefinedWeb (Penedo et al., 2023), a popular pretraining dataset with 960 million documents and 600 billion tokens. The number of documents drastically drops when filtered by length, with those exceeding 64K tokens representing less than 0.1% of the dataset, making it impossible to even construct a 1-billion-token corpus from them. To overcome the data shortage, a common approach is to concatenate documents and split the concatenated sequence into equal-length chunks (Xiong et al., 2024; Chen et al., 2023). The length of each chunk is the same as the target maximum context length. To further enhance data quality, documents longer than a certain threshold are upsampled (Together, 2023). Such a threshold is typically the maximum context length during pretrain. An intuition of this value is that any document longer than this length is chunked during pretrain, so the model has never seen the whole sequence. However, we will later show that such a threshold is far from optimal.

In addition to this, LongAlign (Yushi et al., 2024) shows that before concatenation, documents should be reordered to balance the number of documents within each chunk. In this way, the contribution of each document to the training loss is also balanced. This is orthogonal to this paper and can be directly applied.

**Benchmark** To evaluate the long-context understanding ability, several tasks have been proposed, such as question-answering, code completion, and summarization. For example, Long-Bench (Bai et al., 2024) artificially design questions based on long sequence materials from public datasets (Kočiský et al., 2018; Dasigi et al., 2021); Shaham et al. (2023) introduced sentiment aggregation and reconstructing permuted documents. However, these materials are generally limited to around 10K tokens, far below the maximum context length of concunrret models. Additionally, they are susceptible to contamination as building from public data. The Needle-in-a-Haystack (Kamradt, 2024) offers an alternative. It embeds arbitrary "needle" sentences (e.g. "A is B") into irrelevant long passages, and requires the model to retrieve the needle (e.g. "What is A?"). While this improves scalability and robustness, needle test only evaluates basic retrieval capabilities. To address these issue, recent benchmarks (Hsieh et al., 2024; Zhang et al., 2024) introduce more complicated tests, such as multi-query needle, word frequency counting, etc.

## 3 LONG-DISTANCE REFERRALS IMPROVE DATA QUALITY

Previous studies on long-context training apply a brute-force approach to construct long-context training data corpus. Typically, they select a sequence length threshold of either 4K or 8K tokens. During sampling, documents longer than this threshold are retained, while shorter ones are randomly dropped with a fixed probability. Some recent work (ChatGLM, 2024) proposes a more sophisticated strategy using multiple thresholds. However, there is no experimental evidence that validates the optimality of these thresholds. Instead, given the exponential distribution of document lengths, most sampled documents cluster around the threshold, which takes a significant portion of the token budget. This raises doubts about the efficacy of using a fixed threshold.

Wu et al. (2024) claims that during long-context training, some attention heads develop the ability to capture long-distance dependencies. Building on this idea, we emphasize the importance of long documents, as they provide opportunities for models to learn from long-distance referrals. We design several experiments to prove this hypothesis.

### 3.1 LONG-DISTANCE REFERRAL AND ITS DENSITY

**Definition** We first formally define the long-distance referral. We use $p$ to denote an n-gram, with $s(p)$ the semantic of $p$, $loc(p)$ the index of sentence that $p$ is at. A *referral* $r$ is as a pair of n-grams $p$ meaning the same thing. Formally, $r = (p_1, p_2)$ where $s(p_1) = s(p_2)$. When a concept is mentioned more than twice in the document, we count all pairwise referrals: $(p_1, p_2), (p_1, p_3), (p_1, p_4) \ldots (p_2, p_3) \ldots$. The distance of a referral, $d(r)$ where $r = (p_1, p_2)$, is defined as the number of sentences between the two phrases, i.e. $loc(p_2) - loc(p_1)$. Referrals are divided into several bucket of distances. Given a distance threshold $d_0$, the referral density of a document $doc$ is noted as density$(d_0, doc)$. It is the number of referrals longer than the distance divides the token-level length of the document. This can be formally represented as:

$$\text{density}(d_0, doc) := |\{r|d(r) \geq d_0, r \in doc\}| / \text{length}(doc)$$

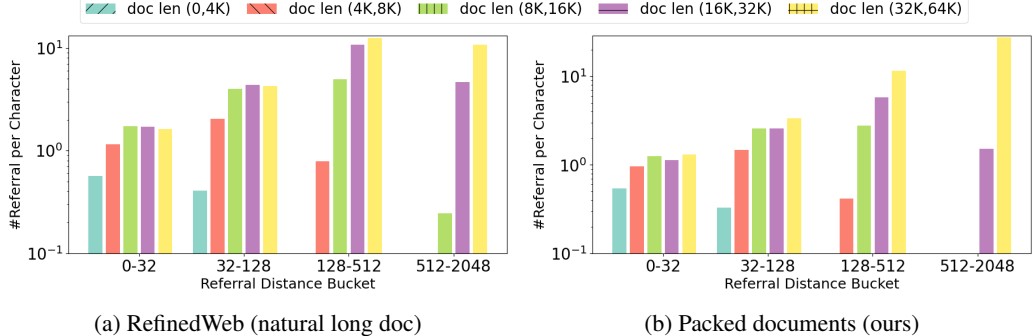

(a) RefinedWeb (natural long doc)  (b) Packed documents (ours)

Figure 1: Density of pairwise referrals in difference sentence-level distance buckets. The document length is measured by the number of tokens.

Likewise, we can define the density regarding a distance bucket $B$.

**Implementation**  For each document, we obtain the top-1000 important phrases based on the text rank of textrank (Nathan, 2016), as well as the named entities (Li et al., 2020) implemented in spaCy (Honnibal et al., 2020). When choosing the top-1000 important phrases, those only contain stop words are skipped. This avoids counting the reference of words like "they", "for", etc.[1] We only count referrals of these phrases. We use coreferee (Hudson & Soumah, 2020) to count not only exact string matches, but also co-references of each phrase. We counts referral distance at the **sentence-level**. On average, a phrase has 2.35 tokens, and a document introduced 253 named entities in addition to the top-1000 important phrases. We divide the sentence-level referral distances into 4 buckets: (0,32), (32,128), (128,512), (512,2048).

Documents are categorized into several groups based on their length, and we conduct referral density analysis across the groups. We set 5 groups of document length intervals: (0-4K), (4K-8K), (8K-16K), (16K-32K), (32K-64K). For each interval, we sample a group of documents within the length interval. Each group has roughly 1 billion tokens. There were insufficient documents even from the whole RefinedWeb dataset to construct a (>64K) group, so we stop at (32K,64K).

**Result.**  Figure 1a plots the results on a 50K sample from the RefinedWeb (Penedo et al., 2023) pretraining dataset. For each referral distance bucket, all documents much longer than it have the same referral density. When the referral distance approaches the document's length, the referral density suddenly drops as expected. In addition to this, simply concatenating short documents do not increase the density of long referrals. Hence, those data chunks originated from long documents have much more long-distance referrals.

We also reported two other metrics in appendix: 1) the density of neighboring referrals within a distance bucket; and 2) the number of concepts which has at least one referral within a distance bucket. The result is in Figure 6. Both metrics are coherent to the pairwise referral density that documents longer than the distance have roughly the same density, but the density drastically drops once the distance is close to the document length.

### 3.2 IMPACT OF REFERRAL DISTANCE AND DENSITY ON DATA QUALITY

To study the effect of referral distance and density, we train the same model on different data sources with the same hyper-parameters. We measure the model's long-context understanding capacity by RULER (Hsieh et al., 2024), a comprehensive long-context ability evaluation framework containing five categories of tasks. It includes multiple previous works such as Needle-In-A-Haystack (Kamradt, 2024), KV-Retrieve, and Multi-QA Liu et al. (2024a).

**Natural documents.**  In the first experiment set, we train the model with natural documents from different length groups. We use the same document length groups as Section. 3.1.

---

[1]We directly use the stop word list from spaCy

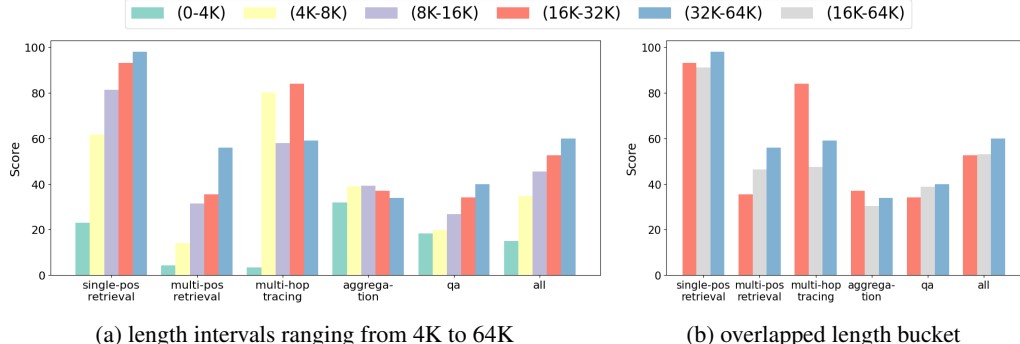

(a) length intervals ranging from 4K to 64K   (b) overlapped length bucket

Figure 2: Model performance when trained on RefinedWeb documents of different length intervals.

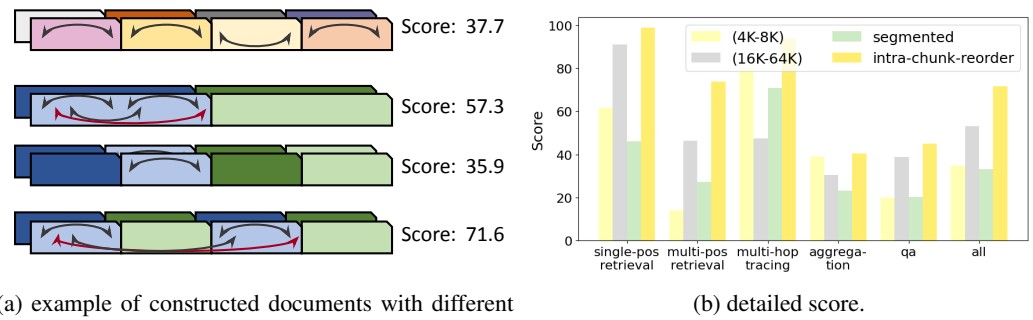

(a) example of constructed documents with different referral length.

(b) detailed score.

Figure 3: By keeping everything but removing long referrals, the data quality (measured by RULER score of model trained with the data) suddenly drops. By en-longing referrals, the data quality drastically increases. Figure 3a Top: data chunks of docs with 4k-8k tokens, which only have short referrals; Second from top: data chunks of docs with 16k-64k tokens, having both long and short referrals; Third from top: data chunks of 4k-token segments from different docs with 16k-64k tokens, where all long referrals (red) are broken; Bottom: data chunks of 4k-token segments from the **same group of documents**, where all long referrals are even longer than before.

The result is visualized in Figure 2a. All document length group longer than 8K achieved high scores on the basic Needle-In-A-Haystack (NIAH) tests (corresponding to the single-pos retrieval category in RULER). However, the NIAH score only measures basic single-location retrieval ability. When assessing a more complicated long-context understanding ability, there is a significant variation across different document length groups.

Generally, although all groups are trained on chunks of the same length, those using longer documents (and thus more longer referrals) to construct chunks performed better in both overall score ("all") and most tasks ("single-pos retrieval", "multi-pos retrieval", "qa"). The performance of "aggregation" does not vary a lot, and document of (4K-8K) and (16K-32K) performs the best for "multi-hop tracing". We reason it as that the task requires to summarize multiple positions of the context, while training with long documents are likely to "lost in the middle" (Liu et al., 2024a).

In addition to this, Figure 2b shows that for tasks related to the source document length, using (16K-64K) result in a performance close to (16K-32K) instead of (32K-64K). Hence, the lower bound of document length will be the bottleneck.

**Constructed documents of different referral density.** In the second experiment set, we ablate the impact of other factors between document groups (e.g. quality, ...). We use exactly the same documents from the (16K,64K) length group for all experiments. In the first experiment, each document is now split into 4K-token segments, and segments from the same document are dispatched into different 128K-token chunks. On the other hand, we conduct experiment to increase the referral distance. In this test, we also split documents into 4K-token segments. Instead of dispatching segments from the same document into different data chunk, we keep them in the original chunk. However, we reorder segments in a chunk in a round-robin pattern: we first have the first segments

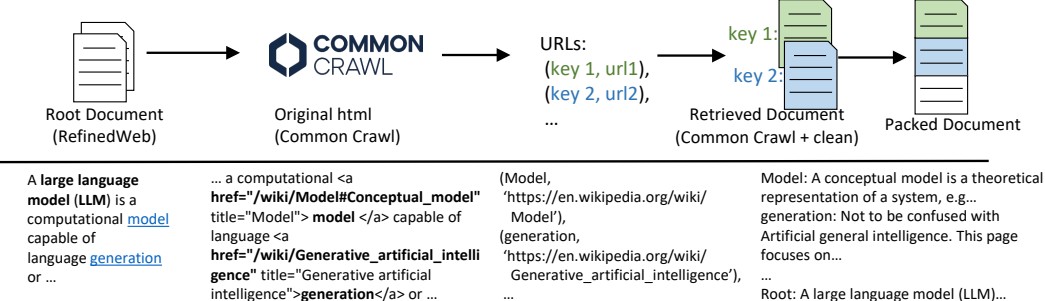

Figure 4: Data construction pipeline in LongPack. From a root page (English Wikipedia of "Large Language Model"), we look up to the html file from Common Crawl, and parse hyper-links in the html as pairs of "key, url" (e.g. key is "generation", url is the Wikipedia page of "generative artificial intelligence"). Then we retrieve and clean contents of the urls, and pack them together as a single long document.

of all documents in this chunk, then the second segments of all documents (if they have the second segment), etc. An example of the two process is in Figure 3a.

By removing the referrals longer than 4K tokens, most referrals shorter than 4K tokens (and thus density$(< d_0, doc)$ with small $d_0$) remains, while those longer than 4K are all broken (density$(\geq d_0, doc) \approx 0$). The overall score suddenly drops 37.6% (from 53.1 to 33.2). This score is at the level of the (4K,8K) group, which now has the same long-referral density.

We name this experiment "intra-chunk-reorder". This method enlarges the distance of all referrals crossing segments, i.e. $d(r)+ = \text{len(segment)} \times \text{num-doc} \times k$, if $r$ crosses $k$ segments in the original chunk. This method drastically improve the overall performance (33.9%), suggesting a strong evidence to the importance of long referral.

## 4 CONSTRUCTING LONG SEQUENCE DATA WITH LONGPACK

Results in Section. 3.2 indicate a clear need for more and longer documents to enhance long-context training. On the other side, Figure 5a demonstrates that long documents are scarce, making it challenging to construct training datasets on the scale of billions of tokens.

To address this issue, we propose LongPack, a data pipeline to construct "near-natural" long documents for long-context training. As proven before, the key attribute of nature long documents is the dense and long-distance referrals. Consequently, the construction of high quality long documents should focus on including rich and long referrals. The constructed long documents is supposed to have a long referral density as close as that of natural documents with the same length.

We further observe that most pretraining data are derived from web pages. For example, RedPajama (Weber et al., 2024), a dataset reproducing Llama training dataset, has 87% from code, and 4.8% from GitHub, which is also available on the Internet These pages are regularly collected by automated crawlers like Common Crawl and Wayback Machine (Crawl, 2024; Archive, 2024). These crawlers capture snapshots of the Internet, storing the content in HTML format, and extracting plain text. Pretraining datasets typically undergo a data cleaning pipeline applied on the plain text. This process removes useless metadata in the HTML, such as the position, font, and size of the text. However, we recognize that hyperlinks present within these web pages are valuable for constructing long documents with embedded referrals.

Hyperlinks often serve two purposes: first, they are used by authors to support the content of a web page, implying a potential causal connection. Second, hyperlinks usually include a brief description or text, which can be seen as a summary or referral to the linked content. We refer to this anchor text as the *key* of the content.

For example, the Wikipedia page on "Large language models" begins with the sentence, "A large language model (LLM) is a computational model capable of language generation or...". To explain the term "generation", the author has inserted a hyperlink to the Wikipedia page for "Generative

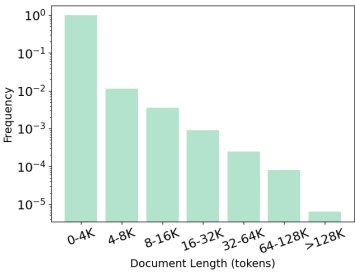
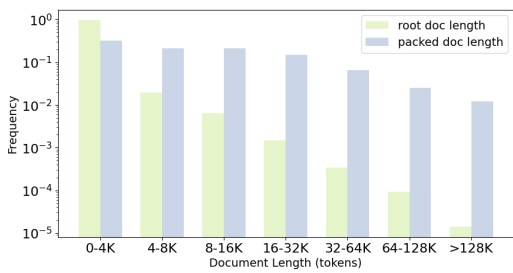

(a) length distribution of RefinedWeb          (b) length of root and packed documents

Figure 5: Document length distribution in our experiments. Root documents are sampled from RefinedWeb, inheriting its exponential distribution. Packed documents are constructed by LongPack based on root documents.

artificial intelligence". In this case, the two pages share a causal relationship, with "generation" serving as the referral key.

Based on this observation, LongPack implements the following process to construct long-sequence data from web pages:

1. We sample web pages from a pretraining dataset, such as RefinedWeb or RedPajama. These sampled pages are referred to as 'root pages'.

2. For each root page, we retrieve the original HTML from Common Crawl and extract all hyperlinks from the HTML. As a result, each root page has a list of (key, URL) pairs.

3. We then use Common Crawl to retrieve the content of the URLs.

4. To ensure data quality, we clean and deduplicate the retrieved pages following methods in pretraining data pipeline (Dodge et al., 2021; Soboleva et al., 2023). Each root page then has a list of (key, content) pairs.

5. Finally, we pack the pages by prepending all retrieved contents before the root document. We order them by the appearance of their hyper-links in the root page. We also prepend its anchoring text on an individual line before the page.

In practice, we extract hyper-links matching all href attributes in the "<a>" tag via a regex expression. The content of the tag is defined as the anchor text of this hyper-link.

To prevent from toxic or low value content, we only get documents from the RefinedWeb. Although this aggressive approach may drop many high-valued pages, we still observe a strong scalability in our experiments. Furthermore, some studies (Hernandez et al., 2022; Muennighoff et al., 2023) prove that data repetition could hurt the model quality. To avoid this issue, when there are multiple root pages referring the same hyper-link, we only keep the hyper-link and its content in the first one.

An example of LongPack and the value extracted at each step is shown in Figure 4.

## 5 EXPERIMENT

In this section, we show that: 1) our method scales in both document length and quality; 2) the constructed document is as good as natural documents at the same length, if not better.

### 5.1 EXPERIMENT SETUP

**Source data** We use the RefinedWeb (Penedo et al., 2023) dataset in all experiments. It is cleaned from several cralws in Common Crawl, which regularly collects snapshots of the whole Internet. The public version of RefinedWeb has 968 million documents and 600 billion tokens.

**Benchmark** We evaluate the model performance on the RULER (Hsieh et al., 2024) benchmark. It contains 4 categories of 13 tasks in total: 1) retrieval. This includes needle-in-a-haystack (niah) and its variants. We split it into two sub-categories; 2) multi-hop tracing, which requires the model to trace a whole chain scattered in a document; 3) token-level aggregation, which counts the frequency

of all words; 4) question answering (QA). For each task, it samples 500 times and uses the pass rate as the score. We report the mean score of each category under a 128K context length, as well as the overall average score. A detailed explanation of each task is in Appendix. C.2. For each training experiment, we report the score of each task in Table 7.

**Model** Due to the limit of computational demand, we use GLM-4-9B (Zeng et al., 2024) as the base pretrained model. It is a state-of-the-art pretrained Transformer model with 9 Billion parameters, and a maximum context length of 8192. GLM-4-9B uses rotary position embedding (RoPE) (Su et al., 2024). Following the practice of previous works, we interpolate the position embedding by making the rotary frequency 16 times larger. Here 16 comes from the ratio of our target maximum context length (128K) to the model's original maximum context length (8K).

Specifically, as we only train for 1 billion tokens, the model's performance cannot exceed the base model's capacity, which is measured by the score with its pretrained context length. We choose the best model at the scale of 10 billion parameters to minimize this factor. However, it still cannot get a perfect score at aggregation and question-answering tasks. We report this score named as "Ideal".

**System** All experiments are running on a single node with 8×H100 80GB GPUs. We use Huggingface Transformers for model implementation, and Huggingface Accelerate for training. For multi-GPU parallelism, we use RingAttention for self-attention to reduce the long sequence activation's memory cost. Other components are parallelized by DeepSpeed-ZeRO.

All experiments share the same hyper-parameters, which are listed in Table 3.

## 5.2 SCALABILITY

Figure 5b briefly shows the length distribution of root documents and the constructed documents. Although looking up to RefinedWeb only data will miss many hyperlinks, there are still 11.7% root records successfully finding at least one content of the hyperlinks. We only counts documents with at least one matching record. In this way, the documents length in average increases 32.7 times (there are some very short pages packed with long referrals), and the total number of tokens (of these 11.7% pages) increases 13 times.

As a result, LongPack can continue scaling from two sides: on one hand, we can increase the number of root documents to scale the data quantity. Given that we only uses 5 million documents, which only contains 0.5% of the whole RefinedWeb dataset, We can scale up the dataset size by an order of magnitude or more; on the other hand, since we introduces approximately we can continue scaling up the length of each document. Although many pages are still limited to 4K-128K after packing, we can do a multi-round packing to generate longer packed documents.

## 5.3 REFERRAL DENSITY

Besides the scalability, we evaluate the quality of documents packed by LongPack, and first look into the referral density.

We first count the referrals and compare them to natural documents with the same length. Figure 1b shows the density of referrals in data constructed by LongPack. With the same density of short distance referrals, our data has a higher density of very long distance referrals, but a slightly lower referral density for short referrals.

Like what we do for natural documents in Section. 3.1, we also reported two other metrics in Appendix. C.1, the neighboring referral density, and the number of concepts with long referrals. For the two metrics, the value on natural long documents has a significant gap from that on short documents. On the other side, our packed documents, even originated from short ones, have the same quality as natural long documents.

## 5.4 END-TO-END PERFORMANCE

To evaluate the performance of models trained on this dataset, we introduce four baselines:

**Domain specific data source.** Following previous practice (Peng et al., 2023; Li et al., 2023), we construct two datasets using PG-19 books and public chat history with ChatGPT. These two datasets

Table 1: Long context understanding performance of model trained on each dataset measured by RULER (↑). Upsampling is the baseline representing previous state-of-the-art data recipe; Ideal refers to an upper bound showing the model's capacity. A more detailed score is in Table 7.

| Name | All | single-pos retrieval | multi-pos retrieval | multi-hop tracing | aggregating | qa |
|---|---|---|---|---|---|---|
| Books | 60.61 | 91.3 | 60.26 | 68.7 | 31.3 | 40.8 |
| Dialogues | 62.15 | 99.0 | 53 | **91** | **38.6** | 38.9 |
| Upsample | 55.1 | 97.5 | 49 | 68.0 | 26.4 | 28.8 |
| Ours | **73.0** | **99.5** | **80.9** | 87.2 | 37.8 | **41.6** |
| Ideal | 89.3 | 99.7 | 97.4 | 99.6 | 78.3 | 59.3 |

are named "Books" and "Dialogues". We do not directly use existing model to control other factors including the base model quality, target context length, and training dataset size.

**Mixed data source.** Domain specific data, although contains many long context data, could drop the model's general quality due to the domain shift. Hence, we also construct a dataset by upsampling long sequence data of each domain in the pretraining dataset. This method is first proposed by Fu et al. (2024). The only difference in our practice is that we set the upsampling threshold to 8K tokens, because that is the initial context length of our base model. Like domain specific data source, we also use the same base model to train the data, because our base model is stronger than that in the paper. We name this case as "Upsampling".

**Upper bound.** Our continuous training dataset only has 1 billion tokens, which is fewer than 0.1% of the pretrain dataset. Hence, we don't expect an improvement on the model's general capability. In other words, the base model's quality will limit the score of some tasks. Hence, we evaluate the base model on its pretrained context length (8K tokens), and report this as "Ideal".

We use "Ours" to denote the score of training with our data. We *do not include* "intra-doc-reorder" in any method because we want to study the data quality without any other effect.

The result of all experiments is in Table 1. Both Books and Dialogues perform better than Upsample. We reason this as that the fact that Upsample does not maintain a long context length. It employees a constant possibility to discard short context documents, while the document length has an exponential distribution. As a result, most documents of the Upsample are at the length of 0-8K (even after a random drop) and 8-16K tokens.

In addition to this, Dialogues perform better than Books on multi-hop tracing and aggregation, but worse at retrievals. We hypothesize that this is because dialogues can be considered as a assembly of multiple turns, each is a relatively independent material. Hence, it performs worse on a single long text understanding, but better at tasks that requires focusing on multiple positions in the text.

Finally, our method outperforms all these baselines. It only performs slightly worse (2-4%) than Dialogues in multi-hop tracing and aggregation tasks, but is better on all other subtasks, and 17-32% over all baselines. We reason this as that documents created by LongPack not only introduced abundant long-distance referrals, which is beneficial to retrieval and qa tasks; but also is assembled by multiple small documents, making it improve on multi-hop tracing and aggregation, which requires focusing on multiple positions in the material.

Although our method outperforms "Upsampling" with a non-trivial improvement, there is still a gap with the upper bound "Ideal". A primary reason is that for a certain task "common-word-extraction", all experiments perform poorly (no more than 2% pass rate), while the upper bound still maintains a 62% pass rate. The mean score for our data in all other tasks reaches 86.2% of the upper bound, while previous methods can only get 71.3% of the upper bound.

In addition to this, following the custom of previous study (Fu et al., 2024), we show the validation loss at each domain before and after long-context training with out data. We prevent the domain shift by uniformly sampling from RefinedWeb, which is a dataset directly used for pre-training. The pretrain dataset of GLM-4-9B is not public, but the authors mention that the dataset contains multilingual resources with a diverse range of domains including web pages, code, papers, etc. Therefore, we use a sample of SlimPajama (Soboleva et al., 2023) as the estimation. Each domain of SlimPajama is sampled with 10 million tokens. We report the validation loss before and after training

Table 2: General performance before and after training measured by validation loss (↓)

| Name | Common Crawl | C4 | Github | Stack-Exchange | Wikipedia | Books | Arxiv |
|---|---|---|---|---|---|---|---|
| Before Training | 3.54 | 3.28 | 5.00 | 3.81 | 4.23 | 3.70 | 3.97 |
| Post Training | 3.50 | 3.31 | 4.71 | 3.82 | 4.05 | 3.57 | 3.90 |

with our data. The result is shown in Table 2. "GLM-4-9B" means the base model performance before training, while "ours" is the validation loss after training with our data. In most domains the validation loss keeps or drops, only with a marginal increase in C4 and Stack-Exchange($< 1\%$). This suggests that continuous training with our data maintains the model's general purpose performance.

## 6 DISCUSSION

There are many different works on data recipe for long-context training. However, they mainly discuss the source to obtain relatively long documents, such as books, dialogues, or a global up-sampling. The fundamental issue, that is the scarce of natural long documents, remains unsolved. In addition, there is no study on how many long documents we need, or how long documents we want. We summarize our contribution as solving the two problems: on one hand, we show that long-distance referral density could be a critical metric for long-context training data; on the other hand, we construct data according to the observation, which packs short documents by referrals.

Despite the success of scaling high quality data, there still exists potential improvements:

First, the scalability is limited as we only search hyper-links in the RefinedWeb. Using more Common Crawl records may significantly hit more hyper-links (currently 10% hit). More advanced hyper-link extraction from the root page also helps. Besides, it worth trying a second round packing, which packs pages referred by documents we obtained in the first round.

Another bottleneck of scalability is the duplicated reference from different pages. We don't observe significant overlap of reference in the 5 million random samples. However, as the data scales up, such conflicts will be more frequent. One solution is to allow limited repetition. Muennighoff et al. (2023) proves that replicating 4 times in pre-training only has negligible impact on the performance.

Second, although hyper-link is a strong reference signal for web pages, it cannot be applied to other data sources. Some data has their domain-specific strong signal, like explicit reference of a file in the code data ("import" in Python, "#include" in C), or citation in the arxiv paper dataset. On the other hand, some data has no strong signal for referral, such as the math or book dataset.

Finally, not all hyper-links are strongly related to the page content. All pages of an organization typically have a hyper-link to the Home page of the organization, which may just be a portal without meaningful information. Filtering out these hyper-links may further improve data quality.

We also discuss the impressive score of "intra-chunk-reorder". Although this trick significant improves the data quality, it has some limitations: First, the segment size cannot be too small, because the model cannot learn at the boundary of segments. Hence, we cannot use a tiny segment size on many short documents. Second, as segmentation increases the distance of cross-segment referrals, it does not increase referral density. As a result, the original document should have dense enough referrals. We leave combining dataset of LongPack with the intra-chunk-reorder a future exploration.

## 7 CONCLUSION

Scaling up the maximum context length of large language model (LLM) meets the shortage of high quality training data. In this work, we show that long distance referrals plays an important role in data quality. Long distance referral is a pair of phrases meaning the same concept, but located far away. Based on the observation, we propose LongPack to construct long documents with the same quality as natural long documents. LongPack packs pages with contents retrieved from hyper-links in a page. Our experiments show that models trained with the packed documents matches the performance of those trained on natural documents of the same length, if not better. Despite of the quality, LongPack also shows the ability to scale on both document length and quantity.

ACKNOWLEDGMENTS

We would like to thank Long Pham, Rajhans Samdani, Yusuf Ozuysal, Vivek Raghunathan, anonymous reviewers, for their insightful feedback. This research is supported by the Semiconductor Research Corporation (SRC) AIHW award 2024AH3210.

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

## A    COMPUTATION REQUIREMENTS FOR LONG SEQUENCE MODELS

Given a model with a hidden size of $h$, number of attention head $n$, MLP intermediate hidden size $i$. Assume it is trained with a sequence length $s$ and batch size $b$, below we approximate the FLOPs it requires by counting the matrix products. Other layers like layernorm are at least one scale lower, so they are negligible.

For QKV projection and output projection in the attention layer, they are mainly matrix products with shape $(b, s, h)$ and $(h, h)$. This means $2bsh^2$ FLOPs for each matrix product, and $8bsh^2$ in total.

For MLP layer, they are matrix products with shape $(b, s, h) \times (h, i)$ or $(b, s, i) \times (i, h)$. So there are $4bshi$ FLOPs in total (2-layer MLP), or $6bshi$ FLOPs if it's the 2-layer gated MLP.

For the self-attention part of the attention layer, there are mainly two matrix products: the first is Query dots Key, which is of shape $(b, n, s, h/n) \times (b, n, s, h/n) \to (b, n, s, s)$, which has $2bhs^2$ FLOPs; the second is the Attention score dots Value, which is of shape $(b, n, s, s) \times (b, n, s, h/n) \to (b, n, s, h/n)$, which has $2bhs^2$ FLOPs. This means $4bhs^2$ FLOPs in total.

For each matrix product operation, during the backward, it's both inputs need to get the gradient. As a result, each matrix product needs two matrix product of the same FLOPs during the backward, meaning the total FLOPs is threefolded.

Now we consider training a LLama-alike model with $B$ tokens, the total FLOPs is:

$$\text{FLOPs}(B, h, i, s)\frac{B}{bs}(\text{iter}) \times 3 \times (8bsh^2 + 6bshi + 4bhs^2) = 6Bh(4h + 3i + 2s)$$

Taking the parameters from Llama-3-8B (h=4096, i=14336), there is

$$\frac{\text{FLOPs}(B, h, i, 8192)}{\text{FLOPs}(B, h, i, 128000)} = \frac{8192 + 43008 + 8192 * 2}{8192 + 43008 + 128000 * 2} = 0.22$$

.

## B    TRAINING HYPERPARAMETERS

Table 3 records all training hyperparameters we use during the training. The setup is mainly inherited from previous studies Fu et al. (2024).

Table 3: Training hyper-parameter in all our experiments.

| parameter | value | name | value |
|---|---|---|---|
| learning rate | 2e-5 | batch size | 64 |
| mixed precision | bf16 | optimizer | AdamW |
| weight decay | 0 | | |

## C  DETAILED EXPERIMENT RESULTS

### C.1  REFERRAL DENSITY

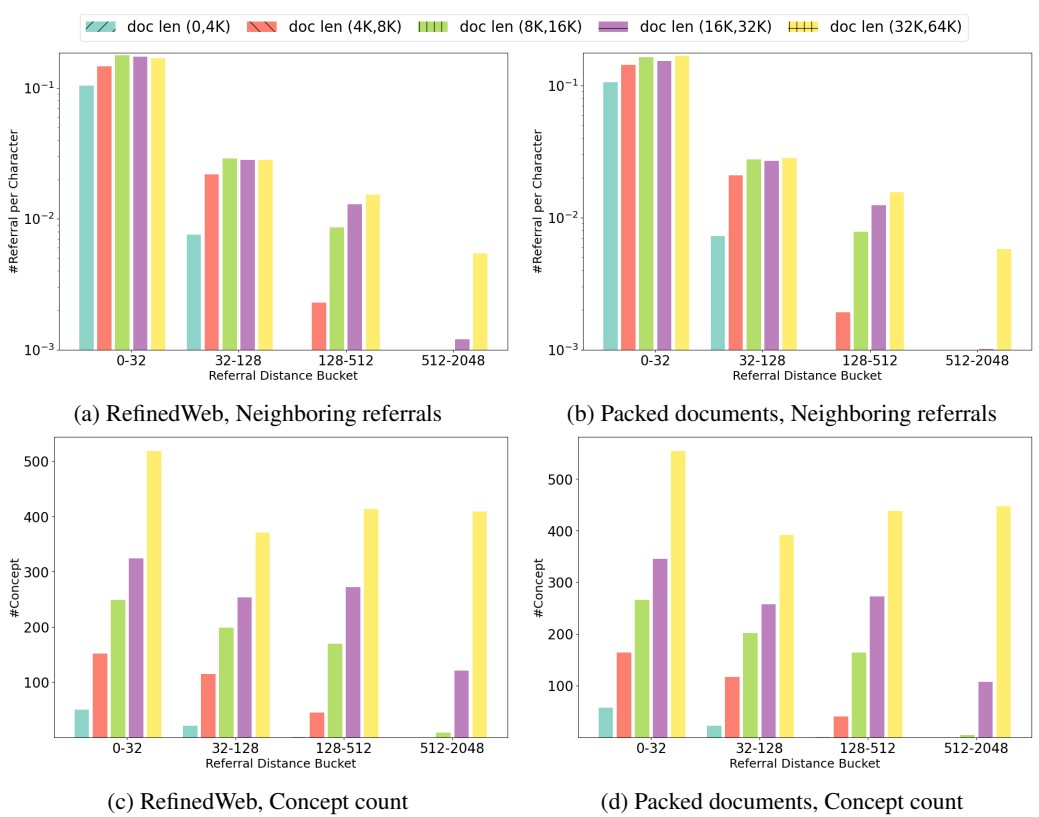

Figure 6: The value of two additional referral metrics: density of neighboring referrals, and count of concept with long referrals.

We first formally define the two new metrics:

**neighboring referral**  Neighboring referrals only counts those pairs of phrases semantically similar, but without a third similar phrase in between them. When a concept has $n$ references in the text, it has $n-1$ neighboring referrals: (0-th,1-th), (1-th,2-th), etc. Like the density of pairwise referrals, we also measure the density of neighboring referrals at each distance bucket. This shows how many referrals are hard to recall when it is mentioned again.

**long-distance referred concept**  This metric counts "how many difference concepts in the text have at least one long referral". Counting on a concept granularity reduces the contribution of words mentioned too many times in the text, and is intended to measure the density of long-distance knowledge within the whole document.

We plot the figure for the two metrics in  Figure 6, and report the exact number of each metrics in  Table 4,  Table 5, and  Table 6, as they look the same on the figure.

Table 4: Density of pairwise referrals within the distance bucket. The value is the density (number of referrals divides number of tokens)

| Name | distance (0-32) | distance (32-128) | distance (128-512) | distance (512-) |
|---|---|---|---|---|
| RefinedWeb (0-4K) | 0.57 | 0.41 | 0.02 | 0 |
| RefinedWeb (4-8K) | 1.15 | 2.03 | 0.79 | 5.08e-03 |
| RefinedWeb (8-16K) | 1.72 | 3.98 | 4.93 | 0.25 |
| RefinedWeb (16-32K) | 1.70 | 4.35 | 10.72 | 4.66 |
| RefinedWeb (32-64K) | 1.62 | 4.26 | 12.57 | 10.73 |
| Packed Doc (0-4K) | 0.54 | 0.33 | 8.23e-03 | 0 |
| Packed Doc (4-8K) | 0.97 | 1.48 | 0.41 | 1.50e-04 |
| Packed Doc (8-16K) | 1.25 | 2.59 | 2.76 | 0.04 |
| Packed Doc (16-32K) | 1.13 | 2.60 | 5.82 | 1.51 |
| Packed Doc (32-64K) | 1.31 | 3.37 | 11.52 | 27.65 |

Table 5: Density of neighboring referrals within the distance bucket.

| Name | distance (0-32) | distance (32-128) | distance (128-512) | distance (512-) |
|---|---|---|---|---|
| RefinedWeb (0-4K) | 0.10 | 7.61e-03 | 8.61e-05 | 0 |
| RefinedWeb (4-8K) | 0.15 | 0.02 | 2.30e-03 | 1.84e-06 |
| RefinedWeb (8-16K) | 0.18 | 0.03 | 8.63e-03 | 9.12e-05 |
| RefinedWeb (16-32K) | 0.17 | 0.03 | 1.29e-02 | 1.20e-03 |
| RefinedWeb (32-64K) | 0.17 | 0.03 | 1.54e-02 | 6.16e-03 |
| Packed Doc (0-4K) | 0.11 | 7.24e-03 | 5.21e-05 | 0 |
| Packed Doc (4-8K) | 0.14 | 0.02 | 1.93e-03 | 2.25e-07 |
| Packed Doc (8-16K) | 0.16 | 0.03 | 7.85e-03 | 3.72e-05 |
| Packed Doc (16-32K) | 0.15 | 0.03 | 1.24e-02 | 1.02e-03 |
| Packed Doc (32-64K) | 0.17 | 0.03 | 1.56e-02 | 6.52e-03 |

## C.2 RULER SCORE OF EACH TASK

In this section, for every model trained in the experiment section, we list the score of each ruler task.

The ruler benchmark has 13 tasks in 4 categories. We further divide the "retrieve" category into two subcategories, single-pos retrieval and multi-pos retrieval. Below we briefly explain each category, as well as the tasks in the category:

**single-pos retrieval** This is a direct mimic of the original Needle-in-a-Haystack (niah). It inserts a single needle sentence and requires the model to find the needle. There are 3 tasks under the category: 1) niah-single-1 (niah-s1) uses repeated noise sentences as the haystack, and the (key,value) pairs are of type (word,number); 2) niah-single-2 (niah-s2) further uses from Paul Graham Essays as the haystack; 3) niah-single-3 (niah-s3) uses Paul Graham Essays as the haystack, and uuids as the value.

**multi-pos retrieval** This is a variant of the niah. There are multiple needles inserted to the haystack. There are 5 tasks: 1) niah-multikey-1 (niah-mk1) has 4 (word) keys with the same (number) value insterted in the Paul Graham Essays Graham (2001–present); 2) niah-multikey-2 (niah-mk2) makes the whole haystack constructed by needles with only one real target; 3) niah-multikey-3 (niah-mk3) also uses distracted needles as the background, but with both key and values being uuids; 4) niah-multivalue (niah-mv) has only a (word) key, but with 4 different (number) values scattered in a Paul Graham Essay; 5) niah-multiquery (niah-mq) has 4 different 4 (word,number) pairs in a Paul Graham Essay.

**multi-hop tracing** This category tests the model's ability to continue tracing a concept along with the whole passage. There is only one task, variable-tracking, which has 4 hops in the whole passage.

Table 6: Counter of long-referral concepts. Concepts are categorized by having at least one referral within the distance bucket. The value is the number of concepts **per document** with at least one referral within the distance bucket.

| Name | distance (0-32) | distance (32-128) | distance (128-512) | distance (512-) |
|---|---|---|---|---|
| RefinedWeb (0-4K) | 49.8 | 20.9 | 0.8 | 0 |
| RefinedWeb (4-8K) | 151.9 | 115.0 | 44.9 | 0.07 |
| RefinedWeb (8-16K) | 248.9 | 198.9 | 169.2 | 8.6 |
| RefinedWeb (16-32K) | 324.0 | 253.8 | 272.0 | 121.4 |
| RefinedWeb (32-64K) | 518.7 | 370.8 | 413.7 | 409.1 |
| Packed Doc (0-4K) | 57.2 | 22.0 | 0.57 | 0 |
| Packed Doc (4-8K) | 164.2 | 116.9 | 40.7 | 0.009 |
| Packed Doc (8-16K) | 266.4 | 201.5 | 164.6 | 4.3 |
| Packed Doc (16-32K) | 345.3 | 257.7 | 273.0 | 107.3 |
| Packed Doc (32-64K) | 554.1 | 391.7 | 437.6 | 447.5 |

**aggregation** This category requires the model to aggregate every position of the passage. There are two tasks: 1) common words extraction(cwe), which asks the top-10 common words in a document; 2) frequent words extraction (fwe), which requires the model to report the top-3 frequent words in the whole passage. Each word is encoded as a uuid.

**question-answering** This category adds distracting information on the normal short context question-answering (QA) tasks. In other words, the question and material are inserted into a noise document. There are two tasks corresponding to two source QA dataset: squad Rajpurkar et al. (2016) (qa1) and hotpotqa Yang et al. (2018) (qa2).

The score of each task for each experiment is listed in the table below. Those trained on RefinedWeb documents at a length interval are named as "RW,{length-interval}". "Segmented" refers to the experiment that splits (16K-64K) documents into 4K segments and dispatches segments to different data chunks. "Upsample" is the baseline of previous upsampling work. "Ours" refers to the model trained on our data. "Ideal" is the ideal performance, which is the model's capability. It is tested on the pretrained maximum context length (8k):

## D  PSEUDO-CODE OF OUR DATA PIPELINE

---

**Algorithm 1** LongPack data process pipeline

---

**Require:** Root document $r$ and its url $u$, HTML database indexed by URL $H$, Cleaned Web Page database indexed by URL $P$

   $html \leftarrow$ look_up$(H, u)$
   $references \leftarrow$ regex_substr_all$(html, REGEX\_REF)$
   $retrieve\_target \leftarrow []$
   **for** $r$ in $references$ **do**
      $url \leftarrow$ regex_substr$(r, REGEX\_URL)$
      $key \leftarrow$ regex_substr$(r, REGEX\_KEY)$
      $retrieve\_target \leftarrow retrieved + [(url, key)]$
   **end for**
   $retrieve\_target \leftarrow$ dedup_url_merge_key$(retrieve\_target)$
   $packed \leftarrow ""$
   **for** $(url, keys)$ in $retrieve\_target$ **do**
      $content \leftarrow$ look_up$(P, url)$
      $packed \leftarrow packed + keys + " : \backslash n" + content$
   **end for**
   $packed \leftarrow packed + "root : \backslash n" + r$

---

The $REGEX\_REF$ is: `<a[^>]+?href="[^>]+?"[^>]*?>[^<]+</a>`

Table 7: Detailed Ruler Score

| Name | niah-s1 | niah-s2 | niah-s3 | niah-mk1 | niah-mk2 | niah-mk3 | niah-mv | niah-mq | vt | cwe | fwe |
|---|---|---|---|---|---|---|---|---|---|---|---|
| RW,(0-4K) | 35.4 | 11.4 | 22.2 | 12.8 | 0.8 | 0.2 | 3.95 | 3.5 | 3.32 | 0.22 | 63.6 |
| RW,(4K-8K) | 100 | 38.4 | 46.6 | 24.8 | 3.2 | 0 | 20.1 | 21.8 | 80.2 | 0.72 | **77.4** |
| RW,(8K-16K) | 100 | 82.4 | 61.6 | 45.4 | 5 | 0 | 53.5 | 53.6 | 57.9 | 1.84 | **76.5** |
| RW,(16K-32K) | 100 | 88.6 | 90.8 | 62.2 | 8.8 | 0.2 | 49.7 | 56.4 | 84.0 | 0.28 | 73.9 |
| RW,(32K-64K) | 100 | 98.6 | 95.6 | 83.8 | 11.5 | 11 | **89.8** | 83.4 | 59.0 | 0.49 | 67.4 |
| RW,(16K-64K) | 100 | 93.2 | 80.2 | 61.8 | 31.8 | 1.6 | 66.1 | 70.5 | 47.4 | 2.7 | 58.1 |
| Segmented | 1 | 72.6 | 64.6 | 52.8 | 4.8 | 0.4 | 41.6 | 36 | 70.76 | 0.9 | 45.6 |
| Intra-chunk reorder | 100 | 98.4 | 98 | 90.8 | 72.4 | 21.6 | **92.5** | **92.3** | **94.0** | **6** | 74.9 |
| Upsample | 95.4 | 98.6 | **98.4** | 80.6 | 9.4 | 9.6 | 69.5 | 75.9 | 68.0 | 1.04 | 51.7 |
| Ours | 100 | 99.8 | **98.6** | **95.8** | **83** | **55** | 86.7 | **84.1** | **87.2** | 0.7 | **74.9** |
| Ideal | 100 | 100 | 99.2 | 96.6 | 98.6 | 95.4 | 97.5 | 99.2 | 99.6 | 62.0 | 94.6 |

| Name | qa1 | qa2 | average | single-pos retrieval | multi-pos retrieval | multi-hop tracing | aggregating | qa |
|---|---|---|---|---|---|---|---|---|
| RW,(0-4K) | 16.2 | 20.4 | 14.9 | 23 | 4.25 | 3.32 | 31.91 | 18.3 |
| RW,(4K-8K) | 18.8 | 21 | 34.8 | 61.7 | 14.0 | 80.2 | 39.1 | 19.9 |
| RW,(8K-16K) | 30.2 | 23.2 | 45.5 | 81.3 | 31.5 | 57.9 | 39.2 | 26.7 |
| RW,(16K-32K) | 44.6 | 23.6 | 52.5 | 93.1 | 35.5 | 84.0 | 37.1 | 34.1 |
| RW,(32K-64K) | 46.5 | 33.5 | 60.0 | 98.1 | 55.9 | 58.98 | 34.0 | 40 |
| RW,(16K-64K) | 47.6 | 30 | 33.2 | 46.1 | 27.1 | 70.8 | 23.2 | 20.1 |
| Segmented | 19.4 | 20.8 | 33.2 | 46.1 | 27.1 | 70.8 | 23.2 | 20.1 |
| Intra-chunk reorder | **51.4** | **38.6** | 71.6 | 98.8 | 73.92 | 94.0 | 40.4 | 45 |
| Upsample | 31.2 | 26.4 | 55.1 | 97.5 | 49 | 68.0 | 26.4 | 28.8 |
| Ours | 45.6 | 37.6 | **73.0** | **99.5** | **80.9** | 87.2 | **37.8** | 41.6 |
| Ideal | 69.6 | 49 | 89.3 | 99.7 | 97.4 | 99.6 | 78.3 | 59.3 |

The $REGEX\_URL$ is: `href="([^">]+?)"`

The $REGEX\_KEY$ is: `>([^<]+)</a>`

