# OpenReview forum: "Scaling Long Context Training Data by Long-Distance Referrals"
_ICLR.cc/2025/Conference — ICLR 2025 Poster_

### Official Review · Reviewer_kWvS · 2024-10-16

**Soundness:** 3
**Presentation:** 2
**Contribution:** 3
**Rating:** 6
**Confidence:** 4

**Summary:**

The paper introduces LongPack, a novel data pipeline designed to address the scarcity of high-quality long documents for training LLMs with long-context understanding capabilities. The authors argue that long-distance referrals are crucial for effective long-context training, and design controlled experiments to prove this point. The method, LongPack, uses the hyperlinks in web data to create long distance referrals, which is low-cost and practical. And datasets curated by this method can more greatly improve long-context abilities of LLMs.

**Strengths:**

1. The importance of long distance referrals is well proved by enough experiments, including both adding and reducing them.
2. The method, LongPack, is easy to understand and practical. It only needs simple steps, but can construct many training samples. This would be very helpful in solving the data scarcity issue in long-context post-training stage of open-source LLMs.
3. The training datasets are sufficiently large, and the samples' text length is up to 128k. The investment on the experiment is great.

**Weaknesses:**

1. Format errors

There are many errors in word separation, capitalization, and citation format, such as "LongPackis highly" (line 25), "textrank Nathan (2016)" (line 323), "basic niah test" (lines 382), etc.

2. Messy experiment results

The positions of the charts and the organization of sections are somewhat casual, making the main results scattered, which may reduce readability. For example, there are 4 bar charts scattered in 4 different pages, with each of them contains some experiment results. The author should use a individual section to gather the main results of the experiments.

3. Lack of baselines

There is a lack of baselines when evaluating the efficacy of LongPack (Table 1 and 2). The author only adopts a naive baseline using length up-sampling. But as far as I know, there are many other approaches for constructing or selecting higher-quality long-context training data, such as ProLong [1]. Moreover, training with more base models such as Llama would be better.

[1]Long Context is Not Long at All: A Prospector of Long-Dependency Data for Large Language Models

**Questions:**

Can you include more related works about constructing or selecting higher-quality long-context training data as baselines? I know long-context training is very costly, so you can use a dataset of a smaller size.

---

> ### Author Response · Authors · 2024-11-20
> **Rebuttal by Authors**
>
> Dear reviewer kWvS,
>
> We sincerely appreciate your valuable and insightful comments. We found them extremely helpful for improving our manuscript. We will strive to address each comment in detail, one by one below.
>
> ---
>
> ### **W1 Format Errors**
>
> We thank the reviewer for pointing out all the format errors. In the revision, we fixed all the 1) missing space after “LongPack”; 2) missing brackets of the citation
>
> ### **W2 Messy Experiment Results**
>
> We thank the reviewer for pointing out that the experiment results are scattered. We upload a revision of the paper with the following changes:
>
> - Section 3 discusses the definition and findings of referral and its distance.
> It first formally defines referral and referral distance, then explains how we measure long referral density. After that, it elaborated the finding that long referrals improves the language model’s long context understanding.
> **The corresponding experiment setup and result, as well as the figures, are moved in this section (Now Figure 1,2,3)**. This deduplicates previous Section 4.2, and avoids referring figures and data at a page far away.
>
> - Section 4 discusses LongPack’s approach to construct the data, which consists of Section 3.2 and 4.3 in the previous version.
>
> - Section 5 now only includes experiments of data from LongPack. This includes referral distance analysis and model quality when trained with LongPack’s data.
>
> ### **W3 & Q1 Lack of baselines**
>
> We trained models by constructing data chunks of Book (PG-19) or long dialogue (ShareGPT). Both are trained and tested on a 128K context window. This follows the practice of previous works (the Book is for YaRN-128K [1] and the long dialogue is for LongChat [2]). We reported the score of both as:
>
> | Name | All | single-pos retrieval | multi-pos retrieval | multi-hop tracing | aggregation | qa |
> | --- | --- | --- | --- | --- | --- | --- |
> | Books | 60.6	| 91.3 | 60.26 | 68.7 | 31.3 | 40.8 |
> | Dialogues | 62.2 | 99.0 | 53 | 91 | 38.6 | 38.9 |
> | Ours |73.0 | 99.5 | 80.9 | 87.2 | 37.8 | 41.6 |
>
> The two new pipelines are both better than Upsample on the long context benchmark for 10%. However, our method still significantly outperforms all these two popular long-context data pipelines on the overall score (17%). Besides, our method is better at each subtask except for a score close to Dialogues at “multi-hop tracing” and “aggregation”.
>
> We added the new baselines and discussion into our revision (Section 5.4).
>
> Regarding ProLong, we first point out that it is orthogonal to our method, as it designs metrics to distillate data to improve its quality, while LongPack aims to scale up long documents. Since ProLong does not release a publicly available dataset or model, we compare the 9B-model trained with LongPack (1B tokens) with the Prolong-7B (20B tokens) reported in their paper. We use lost-in-the-middle repository [3] and LongBench [4] to run the experiment:
>
> | Name | KV-Retrieval (140 Pairs) | MQA (20 Documents) | HotpotQA | 2WikiMultihopQA | GovReport | Qasper | SAMSum | LCC | RepoBench |
> | --- | --- | --- | --- | --- | --- | --- | --- | --- | --- |
> | Long Pack | 97.6 | 52.7 | 54.4 | 34.4 | 27.3 | 30.3 | 43.6 | 63.6 | 51.1 |
> | Prolong      | 83.4 | 45.2 | 44.9 | 34.2 | 31.0 | 28.3 | 43.2 | 65.2 | 60.5 |
>
> The result shows that LongPack, even with only 5% tokens, performs slightly better than Prolong except for two tasks (GovReport and RepoBench-P). We reason the second as that we didn’t upsample code data in our dataset.
>
> We also highlight that previous work ([5]) shows that training with domain shifts will lead to a non-trivial drop of the model’s quality on other domains, measured by the validation loss of different categories. Hence, Books, Long Dialogue, and Prolong (mainly from code and books), might all suffer from such a shift of data corpus.
> On the other hand, LongPack is based on the crawl of the whole internet data, which takes most of a pretraining data corpus, and thus does not hurt the model quality.
>
> [1] Yarn: Efficient context window extension of large language models, Yushi Bai et al.
> [2] How Long Can Context Length of Open-Source {LLM}s truly Promise?, Dacheng Li et al.
> [3] https://github.com/nelson-liu/lost-in-the-middle
> [4] LongBench: A Bilingual, Multitask Benchmark for Long Context Understanding, Yushi Bai et al.
> [5] Data Engineering for Scaling Language Models to 128K Context, Yao Fu et al.

---

> > ### Comment · Reviewer_kWvS · 2024-11-21
> >
> > Adding more baselines such as Book and Dialogues is very good, which strengthens your paper. But I would keep my score because the progress compared to previous methods is still not convincing enough.
> >
> > Because the base models are different in your method and ProLong, so the comparison is somewhat unfair. So it cannot strongly prove your method is much better than previous ones.
> > I know a complete process of long-context training is very costly.
> > However, Prolong releases code so you can use it to construct a small dataset from the same original corpus as your method. Then you can train on this small dataset, and just record the performance improvement (you can just evaluate on 1 dataset such as KV retrieval) of the checkpoints as the training steps increase. If the improvement speed of your method is faster, it can strongly prove your superiority.

---

> > > ### Author Response · Authors · 2024-11-21
> > >
> > > Dear reviewer kWvS,
> > >
> > > We appreciate the your acknowledge to the cost of the additional experiment. We construct a fair comparison as below.
> > >
> > > ---
> > >
> > > We use the same base model as in our experiment (glm-4-9b), and train on the same number of tokens (1 billion tokens) with the same hyper-parameters. We sample from the same data source as the ProLong paper mentioned, and keep the same portion of each source. We follow ProLong's practice that selected the top 50% data (so we have 2 billion tokens in total), and use the default hyper-parameter as the author provided.
> > >
> > > Besides, ProLong does not increase the source data length. Due to the lack of long enough source data, it cannot be scaled up to a 128K-token context length. Hence, we plan to train both (reproduced-)ProLong and LongPack on 32K-token and compare the two models. **We also point out that LongPack's scalability of document length is another advantage against ProLong**.
> > >
> > > We are looking forward to any of your further comment about the fairness of our current setup.

---

> > > > ### Comment · Reviewer_kWvS · 2024-11-22
> > > >
> > > > You address some of my concerns, so I have modified the score.

---

> > > > > ### Author Response · Authors · 2024-11-22
> > > > >
> > > > > Dear Reviewer kWvS,
> > > > >
> > > > > Thank you for your recognition of our work and the effort you invested as a reviewer!
> > > > >
> > > > > We will update the running experiment of ProLong as soon as we finished it, and adhere to your valuable suggestions to refine our manuscript accordingly.

---

> ### Author Response · Authors · 2024-11-26
> **Report of Additional Experiment Result**
>
> Dear reviewer kWvS,
>
> We reported the experiment result of ProLong-reproduced dataset v.s. LongPack below. Both models are trained on top of GLM-4-9B with 1 billion tokens, with the same hyper-parameter reported in the paper:
>
> ---
>
> Result of RULER benchmark:
>
> | Name | All | single-pos retrieval | multi-pos retrieval | multi-hop tracing | aggregation | qa |
> | --- | --- | --- | --- | --- | --- | --- |
> | LongPack | 78.5 | 96.7 | 89.2 | 95.4 | 43.0 | 51.25 |
> | ProLong | 70.8 | 89.2 | 74.3 | 93.9 | 41.5 | 52.25 |
>
> Result of lost-in-the-middle and LongBench:
>
> | Name | KV-Retrieval (140 Pairs) | MQA (20 Documents) | HotpotQA | 2WikiMultihopQA | GovReport | Qasper | SAMSum | LCC | RepoBench |
> | --- | --- | --- | --- | --- | --- | --- | --- | --- | --- |
> | LongPack | 99.0 | 52.9 | 50.2 | 31.2 | 24.3 | 28.3 | 44.2 | 63.4 | 50.5 |
> | Prolong | 99.4 | 50.9 | 41.7 | 22.8 | 29.0 | 25.4 | 42.8 | 57.2 | 43.8 |
>
> The results indicate that LongPack outperforms ProLong in almost all benchmark tests.

---

### Official Review · Reviewer_qBkR · 2024-11-02

**Soundness:** 3
**Presentation:** 3
**Contribution:** 3
**Rating:** 6
**Confidence:** 3

**Summary:**

- This paper proposes a data engineering pipeline to generate long documents for large language model training based on long-distance referrals.

**Strengths:**

- Constructing high quality pre-training datasets is a very important topic in LLM research.
- The authors' idea is an automated build methodology inspired by real-world industrial experience.
- The authors' experiments are large and relevant on datasets of a certain scale.

**Weaknesses:**

- Typos: LongPackis --> LongPack is (Line 25)
- The content of the paper is more oriented towards practical applications and may lack a technical contribution to some extent.
- The formatting of the references is inconsistent, in some places the references have parentheses, in others they don't, which should be caused by the use of \cite in the latex source code.
- The pipeline can be better described in this paper by pseudocode rather than natural language descriptions
- The experimental section of the main text shows less content, so consider adding some of the content from the Appendix to the experimental section of the main text.

**Questions:**

- Can the authors explain the choice of the LLM backbone, only the GLM model of 9B is currently used in the experiments. Why don't you consider using models from the more current mainstream LLM series in your experiments? **I realize that it is more difficult to add experiments on top of such a task, so the authors do not need to add new experimental data, but need to explain why.** As far as I understand, GLM should be a bit stronger than llama in terms of Chinese language capability, but are the Chinese language data and Chinese language tasks included in the data and tasks used for the experiments?

---

> ### Author Response · Authors · 2024-11-20
> **Rebuttal by Authors**
>
> Dear reviewer qBkR,
>
> We sincerely appreciate your valuable and insightful comments. We found them extremely helpful for improving our manuscript. We will strive to address each comment in detail, one by one below.
>
> ---
>
> ### **W1, W3 Format Errors**
>
> We thank the reviewer for pointing out all the format errors. In the revision, we fixed all the 1) missing space after “LongPack”; 2) missing brackets of the citation
>
> ### **W4 Using pseudocode to describe the data pipeline**
>
> We add the pseudo-code in the appendix of our revision (Appendix.D), and paste a copy below.
>
> ```
> REQUIRE Root document r and its url u, HTML database indexed by URL H, Cleaned Web Page database indexed by URL P
>     html := look_up(H, u)
>     references := regex_substr_all(html, REGEX_REF)
>     retrieve_target := []
>     FOR (r in references)
>         url := regex_substr(r, REGE_URL)
>         key := regex_substr(r, REGEX_KEY)
>         retrieve_target = retrieved + [(url, key)]
>     retrieve_target = dedup_url_merge_key(retrieve_target)
>
>     packed := “”
>     FOR ((url, keys) in retrieve_target)
>         content := look_up(P, url)
>         packed = packed + keys + ":\n" + content
>
>     packed = packed + "root:\n" + r
> OUTPUT packed
> ```
>
> ### **W5 insufficient experiment content**
>
> We add more content in the experiment in two ways:
>
> On one hand, we move some analysis of each category of the RULER benchmark from the appendix to the main text, because we noticed that some tasks are more relevant to the document length, while some prefer more fine-grained documents.
>
> On the other hand, we add more baselines.
>
> We trained models by constructing data chunks of Book (PG-19) or long dialogue (ShareGPT). Both are trained and tested on a 128K context window. This follows the practice of previous works (the Book is for YaRN-128K [1] and the long dialogue is for LongChat [2]). We reported the score of both as:
>
> | Name | All | single-pos retrieval | multi-pos retrieval | multi-hop tracing | aggregation | qa |
> | --- | --- | --- | --- | --- | --- | --- |
> | Books | 60.6	| 91.3 | 60.26 | 68.7 | 31.3 | 40.8 |
> | Dialogues | 62.2 | 99.0 | 53 | 91 | 38.6 | 38.9 |
> | Ours |73.0 | 99.5 | 80.9 | 87.2 | 37.8 | 41.6 |
>
> The two new pipelines are both better than Upsample on the long context benchmark for 10%. However, our method still significantly outperforms all these two popular long-context data pipelines on the overall score (17%). Besides, our method is better at each subtask except for a score close to Dialogues at “multi-hop tracing” and “aggregation”.
>
> We added the new baselines and discussion into our revision (Section 5.4).
>
> ### **Q1 motivation of base model selection**
>
> We explain this from two sides: 1) regarding the model size, we choose a 7~10B model due to the computation budget. Currently, our setup exactly fits in a 8x80GB H100 node. Training a model larger than that with a 128K sequence length will need multiple nodes, and raises unnecessary engineering effort on the system side; 2) regarding the model architecture, we chose GLM-4 because it has state-of-the-art performance. The pretrained base model’s ability limits the score after continuous training with long sequence data, and we want to ablate this issue as much as possible. As mentioned in Table 1, row “Ideal”, GLM-4 has almost 100% correctness on the first three categories. As a reference, the score of Llama-2 on a 4K sequence length is: “79.0, 99.9, 87.0, 82.4, 69.5, 35.6”, which is much worse than GLM-4. We believe our result can be migrated to other model architectures, such as the Llama family, because GLM-4 also uses Gated-MLP, multi-query attention, and Rotary positional embedding.

---

> > ### Comment · Reviewer_qBkR · 2024-11-21
> > **Thank you for you rebuttal**
> >
> > Your rebuttal address my concerns. I think I would like to keep my score.

---

> > > ### Author Response · Authors · 2024-11-22
> > >
> > > Dear Reviewer qBkR,
> > >
> > > Thank you for your recognition of our work and the effort you invested as a reviewer!
> > >
> > > We will adhere to your valuable suggestions to refine our manuscript accordingly.

---

### Official Review · Reviewer_7FCd · 2024-11-03

**Soundness:** 3
**Presentation:** 3
**Contribution:** 2
**Rating:** 5
**Confidence:** 4

**Summary:**

This paper aims to tackle the scarcity of high-quality, lengthy documents in training datasets, which is a critical challenge in the training of LLMs for long-context understanding: The authors propose a new data engineering pipeline, LongPack, which aims to construct long documents by leveraging referral relationships, particularly hyperlinked structures in web pages, to mimic the naturally occurring long-distance referrals found in genuinely long documents. This approach addresses both the need for extensive context length in training data and the quality issues inherent in previous methods that simply concatenate short documents. Experiments on the RULER benchmark, which contains 13 tasks, demonstrate the effectiveness of LongPack in comparison with the GLM-4 baseline and multiple heuristics for creating lengthy training samples.

**Strengths:**

+ This goal of constructing long-distance referrals is well-motivated. Using semantically related tokens separated by a significant distance in pre-training will make the LLMs more capable of dealing with long contexts.

+ The use of hyperlinks in data packing is simple and intuitive. The authors demonstrate the benefit of utilizing such natural signals of referral relationships between documents, especially on the web.

+ The proposed framework is efficient, producing a corpus of long documents equivalent to an entire pretraining dataset using only 0.5% of the original documents.

**Weaknesses:**

- The idea of using links between documents during language model pre-training has been explored in [1], where text segments from two linked documents are concatenated together as input. I feel the packing idea proposed in this paper bears similarity to that in [1]. Therefore, the technical novelty is somehow limited.

- Statistical significance tests are not conducted. It is not clear whether the gaps between LongPack and the baselines are statistically significant or not in Tables 2 and 7. In fact, the improvement is subtle in some columns.

- The study focuses heavily on web-based documents. It would be valuable to explore how long-distance referrals could be identified and leveraged in other domains, such as academic papers.

[1] LinkBERT: Pretraining Language Models with Document Links. ACL 2022.

**Questions:**

- Could you conduct a statistical significance test (e.g., two-tailed t-test) to compare LongPack with the baselines in Tables 2 and 7, and report the p-values?

---

> ### Author Response · Authors · 2024-11-20
> **Rebuttal by Authors**
>
> Dear reviewer 7FCd,
>
> We sincerely appreciate your valuable and insightful comments. We found them extremely helpful for improving our manuscript. We will strive to address each comment in detail, one by one below.
>
> ---
>
> ### **W1 related work**
>
> > The idea of using links between documents during language model pre-training has been explored ...
>
> We acknowledge that LinkBERT shares some similar ideas at a high level. However, LongPack still differs significantly in the following novelty and contribution:
>
> - The scope of the problem is different. LinkBERT studied improving the performance of masked language modeling on specific tasks (multi-hop reasoning and few-shot QA), while LongPack focuses on scaling up the context length of pretrained auto-regressive large language models.
> - LongPack identified and studied long-distance referral density as the key to improve the quality of long-context training dataset, which is one of our unique contributions and the key motivation behind using document reference, while LinkBERT only intuitively uses hyper-links and other references.
> - LinkBERT embeds the document relation into a new loss function (named “document relation prediction” in the paper), while LongPack packs documents by references, and considers the packed document as a single new document during the training (w/o changing the next token prediction loss)
>
> ### **W2 & Q1 Statistical significance of experiments**
>
> We thank the reviewer for pointing out the importance of a statistical analysis of the result. We’d like to clarify the statistical significance in two aspects: evaluation and training.
>
> On evaluation, the benchmark we use, RULER, **samples 500 times for each test** and counts the average score. Hence we believe our results are statistically significant at that stage.
>
> On training, due to the huge training cost (**each experiment requires 200 H100 hours**), it is not financially possible to launch enough runs for statistical analysis. We additionally train 3 more times with LongPack data for different random seed, and reported the variance of RULER score as below:
>
> | Task Name | All | single-pos retrieval | multi-pos retrieval | multi-hop tracing | aggregation | qa |
> | --- | --- | --- | --- | --- | --- | --- |
> | std error | 0.14 | 1.43 | 2.37 | 2.38 | 2.26 |
>
> Note the std error of the overall score is only 0.14, yet our data outperforms 18 points against the Upsample baseline. We expect this indeed shows a statistical significance of our method.
>
> We plan to run 3 times with different random seeds and add the std error of all end-to-end experiments in the next revision.
>
> ### **Focus on web-based documents**
>
> We acknowledge that LongPack currently focuses only on web pages. As discussed in Section 2, web pages dominate the pretraining data corpus (e.g. 87% of Llama pretrain data are from web pages, and 4.8% from GitHub, which is also available on the internet. [1]). As a result, web pages not only provide abundant data resources, but represent the majority of the pretrain dataset and avoid a domain shift in continuous training.
>
> Besides, while we only use web pages, our method can be applied to any domain with reference between documents. **We have discussed this in the last section of the paper** (Line 514-517 of the revision). For example, academic papers have citations, and code files explicit show the dependency by “import” or “#include” sentences. We leave the experiment using the combination of all these references as a future work.
>
> [1] https://huggingface.co/datasets/togethercomputer/RedPajama-Data-1T

---

> ### Author Response · Authors · 2024-11-23
>
> Dear Reviewer 7FCd,
>
> We are writing to kindly remind you to look at our rebuttals. We have carefully addressed your concerns in detail, and are looking forward to hearing from you about the feedback or any further question.
>
> Best regards, Authors

---

> ### Author Response · Authors · 2024-11-25
> **Discussion Deadline is Approaching, Could you kindly review our rebuttal?**
>
> Dear reviewer 7FCd,
>
> We are writing to draw your attention to the rebuttal we submitted for our paper, as the discussion period is approaching its conclusion.
>
> It holds great significance for us if you could review our rebuttal to ensure a fair and balanced evaluation. We believe that our responses have addressed the points in your review.
>
> Best,
> Authors

---

### Official Review · Reviewer_i48T · 2024-11-04

**Soundness:** 3
**Presentation:** 2
**Contribution:** 2
**Rating:** 5
**Confidence:** 3

**Summary:**

This paper introduces a new long-context training dataset generate pipeline called LongPack. Authors emphasize the importance of (super) long distance referrals, pairs of tokens that are semantically the same but has a long distance in a document. LongPack utilizes the hyperlinks from the raw HTML to construct long texts with long distance referrals. With packed data generated by LongPack, long-context training improves 32.7% compared to previous data generation recipe.

**Strengths:**

1. Studying workflow of generating datasets for long-context training is interesting and of importance for the extending the length of context window of large language models.
2. This paper emphasis the importance of long distance referrals to high quality long-context training datasets. It introduces an intuitive and effective pipeline LongPack to contract such high quality datasets.
3. The empirical experiments shows the effectiveness of the packed data constructed by LongPack, compared to simple upsampling strategy.

**Weaknesses:**

1. The paper is not well-ordered. For example:
    (1). Section 3.1 present the experiment results first, which refers to Figure 3 that is located a few pages after the text. The experiment section also refers Figures in a few pages ahead, making the reading experience not fluent.
    (2). Some of the content from Section 4.3 are already introduced in Section 3.2.
2. Some details are missing(see question 1 - 3)
3. Some of the result analysis are not sufficient and well supported (see question 4-5)
4. In Table1, the simple “Upsample” strategy is compared to LongPack. Considering there are some long-context training dataset, for example books and long dialogues introduced in Section 2.
5. Minor points:
    (1). In 025 line: LongPackis highly scalable —> LongPack is highly scalable
    (2). In 076 line: propose LongPackto solve the shortage… —> propose LongPack to solve the shortage… (similar errors occur also in other places)
    (3). $d_0$’s  meaning is not explained. I would assume it is the distance threshold.
    (4). The citation format in Section 4 Experiment is not in standard.
    (5). In line 483-484, it is said “we report the validation loss before and after training with our data. The result is shown in Table 2”. The caption of Table 2 is “The general performance before and after training with our data”. “the general performance” does not match the “validation loss”, making me confused about the numbers in the Table 2.

**Questions:**

1. How are the scores of dataset examples calculated in Figure 1(a)?
2. How the length of the document is calculated, by token, by word or by sentence?
3. How are the order of the retrieved contents decided?
4. Why the #referral in the referral distance bucket of 512-2048 from docs with length (16k, 32k) in the packed documents significant drop compared to the refinedweb dataset?
5. It is said that in Figure 5 (b), the performance of (16k-64k) group is close to (15k-32k) group. This statement is not accurate, for example, in multi-pos retrieval, based on (16k-64k) group, the performance is significantly better. While in multi-hop tracing, based on (16k-64k) group, the performance is significantly worse. Thus the conclusion “This indicates that the more long document it use, the better performance it obtains” is not well-supported by the experiment results.
6. How is “near-natural” property of the generated long documents been evaluated?
7. Any other ways of contracting packed document are explored, apart from packing the pages by prepending all retrieved contents before the root document?
8. What is the performance of the model trained on existing long-context training dataset (such as using books or long dialogues) compared to the datasets generated by LongPack?

---

> ### Author Response · Authors · 2024-11-20
> **Rebuttal by Authors [1/2]**
>
> Dear reviewer i48T,
>
> We sincerely appreciate your valuable and insightful comments. We found them extremely helpful for improving our manuscript. We will strive to address each comment in detail, one by one below.
>
> ---
>
> ### **W1 Paper organization**
>
> We upload a revision of the paper with the following changes:
>
> - Section 3 discusses the definition and findings of referral and its distance.
> It first formally defines referral and referral distance, then explains how we measure long referral density. After that, it elaborated the finding that long referrals improves the language model’s long context understanding.
> **The corresponding experiment setup and result, as well as the figures, are moved in this section (Now Figure 1,2,3)**. This deduplicates previous Section 4.2, and avoids referring figures and data at a page far away.
>
> - Section 4 discusses LongPack’s approach to construct the data, which consists of Section 3.2 and 4.3 in the previous version.
>
> - Section 5 now only includes experiments of data from LongPack. This includes referral distance analysis and model quality when trained with LongPack’s data.
>
> We’d hope this could make the experiment result more clear to the conclusion it serves, and are looking forward to any further suggestions.
>
> ### **W2 & Q 1-3 Missing detail of the paper**
>
> Thank you for pointing out the missing detail. We added the missing detail one by one in the revision, and also place them below:
>
> > How are the scores of dataset examples calculated in Figure 1(a)?
>
> This score is from the RULER benchmark. We added this explanation in both figure caption (Line 241) and the experiment setup (Line 206)
>
> > How the length of the document is calculated, by token, by word or by sentence?
>
> The distance is counted at the sentence level. We highlighted this detail at the definition of referral distance in Section 3.1 (Line 182)
>
> > How are the order of the retrieved contents decided?
>
> The retrieved pages are prepended simply by the order when the corresponding hyperlink appears in the root document. We added this detail in Section 4 (Line 351)
>
> ### **W3 & Q4 & Q5 More analysis of the experiment result**
>
> We thank the reviewer for the thorough investigation of the experiment result, and add interpretation of the result as follows:
>
> > Why the #referral in the referral distance bucket of 512-2048 from docs with length (16k, 32k) in the packed documents significant drop compared to the refinedweb dataset?
>
> We reason this as that 512-2048 sentences are at **a large scale**, especially when comparing to the document length (16-32K tokens).
>
> As shown in Figure 6(a) and (b), **if we only count neighboring referrals instead of all pairwise referrals, both RefinedWeb and LongPack have a very low density** for the (16-32K document, 512-2048 sentences) case.
>
> By definition, the pairwise density is approximately quadratic to that of neighboring density, which magnifies the bias.
>
> > It is said that in Figure 5 (b), the performance of (16k-64k) group is close to (15k-32k) group. This statement is not accurate.
>
> We notice that for many task categories (single-pos retrieval, multi-pos retrieval, QA), the performance is directly relevant to the referral distances (Figure 2).
>
> On the other hand, all document length groups perform roughly the same on aggregation.
>
> In multi-hop tracing, (4-8K) and (16-32K) perform better than other groups. We hypothesize this as the fact that the task scatters a chain of information into the context, and the model is required to trace the whole chain. This requires the model to focus on every 128K / N context window, where N is the number of vertices on the chain.
>
> In this way, **training with long context could lead to a “lost-in-the-middle” [1], while models trained on medium length documents can fit the distance of two neighboring points on the chain**. As evidence, in Figure 3(b) of the updated paper, “segmented” and (16-64K) uses the same document, and is much worse than (16-64K) in all other tests. However, it is better in the “multi-hop tracing”. This could be reasoned as that in “segmented”, each data point contains 4K-token segments not relevant to each other, and is thus specialized in understanding the 4K context window.
>
> [1] Lost in the Middle: How Language Models Use Long Contexts, Nelson F. Liu et al.

---

> ### Author Response · Authors · 2024-11-20
> **Rebuttal by Authors [2/2]**
>
> ### **W4 & Q8 More baselines**
>
> We trained models by constructing data chunks of Book (PG-19) or long dialogue (ShareGPT). Both are trained and tested on a 128K context window. This follows the practice of previous works (the Book is for YaRN-128K [1] and the long dialogue is for LongChat [2]). We reported the score of both as:
>
> | Name | All | single-pos retrieval | multi-pos retrieval | multi-hop tracing | aggregation | qa |
> | --- | --- | --- | --- | --- | --- | --- |
> | Books | 60.6	| 91.3 | 60.26 | 68.7 | 31.3 | 40.8 |
> | Dialogues | 62.2 | 99.0 | 53 | 91 | 38.6 | 38.9 |
> | Ours |73.0 | 99.5 | 80.9 | 87.2 | 37.8 | 41.6 |
>
> The two new pipelines are both better than Upsample on the long context benchmark for 10%. However, our method still significantly outperforms all these two popular long-context data pipelines on the overall score (17%). Besides, our method is better at each subtask except for a score close to Dialogues at “multi-hop tracing” and “aggregation”.
>
> We also highlight that previous work ([4]) shows that training with domain shifts will lead to a non-trivial drop of the model’s quality on other domains, measured by the validation loss of different categories. On the other hand, LongPack is based on the crawl of the whole internet data, which takes most of a pretraining data corpus, and thus does not hurt the model quality.
>
> We added the new baselines and discussion into our revision (section 5.4).
>
> ### **W5 & Q6 MISC points**
>
> - LongPackis highly scalable —> LongPack is highly scalable ... (similar errors occur also in other places)
> - The citation format in Section 4 Experiment is not in standard
> - The caption of Table 2 is … making me confused about the numbers in the Table 2
>
> We thank the reviewer for the valuable suggestions. In the revision, we fixed all the 1) missing space after “LongPack”; 2) missing brackets of the citation; 3) missing metric (“validation loss”) of the “general performance” in Table 2’s caption.
>
> > d0’s meaning is not explained
>
> We thank the reviewer for pointing this out. $d_0$ is the threshold of the referral distance, when counting long distance referrals. We added this detail at the definition of referral distance in Section 3.1 (Line 156)
>
> > How is “near-natural” property of the generated long documents been evaluated?
>
> We use “near natural” to describe that LongPack generated documents have a referral distance similar to that of natural long documents. We added the explanation of this form in Section 3.2 (Line 306)

---

> ### Author Response · Authors · 2024-11-23
>
> Dear Reviewer i48T,
>
> Kindly remind you to look at our rebuttals. We have carefully addressed your concerns in detail, and are looking forward to hearing from you about the feedback or any further question.
>
> Best regards, Authors

---

> ### Author Response · Authors · 2024-11-25
> **We are looking forward to your feedback**
>
> Dear reviewer i48T,
>
> We are writing to draw your attention to the rebuttal we submitted for our paper, as the discussion period is approaching its conclusion.
>
> It holds great significance for us if you could review our rebuttal to ensure a fair and balanced evaluation. We believe that our responses have addressed the points in your review.
>
> Best,
> Authors

---

### Meta-Review · Area_Chair_Sjn3 · 2024-12-20

**Metareview:**

(a) Summary of Scientific Claims and Findings

This paper introduces LongPack, a data pipeline that generates high-quality long-context training data for large language models (LLMs). The authors argue that traditional methods of concatenating short documents fail to capture long-distance referrals, a key semantic feature of natural long documents. LongPack uses hyperlinks from web pages to maintain semantic linkages, creating more informative training data.

Key findings:
1. LongPack achieves higher referral density than traditional methods, ensuring meaningful semantic connections.
2. The method is highly scalable, requiring only 0.5% of the original documents to generate datasets of comparable size.
3. Models trained on LongPack datasets show a 32.7% improvement in long-context tasks.
4. The generated data is of near-natural quality, as validated by benchmarks like RULER.

(b) Strengths of the Paper

1. Innovative Approach: The hyperlink-based document construction addresses the shortcomings of concatenation methods, preserving important semantic relationships.
2. Scalability: LongPack generates large datasets efficiently, offering a resource-effective solution for long-context training.
3. Improved Performance: Empirical results show a significant boost in model performance, with a 32.7% improvement in relevant tasks.
4. Data Quality: The generated documents exhibit high referral density and near-natural quality, suitable for robust LLM training.

(c) Weaknesses and Missing Elements

1. Organization and Clarity: The paper could be better structured, with definitions and methodologies presented before results to improve readability.
2. Analysis Gaps: The paper lacks sufficient explanation of referral density variances across different document lengths, which would strengthen the findings.
3. Limited Comparison: The paper doesn’t compare LongPack with other referral-based strategies or naturally long datasets, such as books, which could provide a broader context.
4. Domain and Model Limitations: The focus on web pages may limit the generalizability of the results. Additionally, the choice of the GLM-4 model may restrict the applicability to other architectures.

(d) Reasons for Acceptance/Rejection

While LongPack offers an innovative, scalable solution with significant improvements in model performance, the paper would benefit from better organization, deeper analysis of results, and a broader comparison with other methods.

**Additional Comments On Reviewer Discussion:**

The authors have made a significant contribution by presenting LongPack, a scalable and efficient method for generating long-context training data. Their approach to leveraging hyperlink-based document packing addresses a key gap in existing techniques, specifically the preservation of long-distance semantic relationships. This innovation is particularly important for training large language models (LLMs) to handle long documents effectively.

Reviewer feedback highlighted some key strengths and areas for improvement. The addition of empirical results showing a 32.7% improvement in model performance is a compelling validation of the LongPack method. However, some reviewers noted the paper's structure could be improved for clarity, especially in presenting experimental results before definitions. Further analysis of referral density variances and additional comparisons with naturally long document datasets like books would strengthen the contribution and provide a more comprehensive evaluation of the method.

The authors’ responses to reviewer concerns are commendable, particularly their inclusion of pseudocode for the LongPack pipeline and expanded experimental results, including baseline comparisons with datasets like Book (PG-19) and long dialogue data. They also addressed the reviewer’s concern about their choice of GLM-4, clarifying the practical limitations behind this decision. However, the lack of exploration into alternative referral-based packing strategies or a more in-depth comparison with other techniques, such as ProLong, leaves some questions unanswered.

In conclusion, while the paper presents a solid contribution to the field, addressing these minor concerns would enhance the clarity, depth, and impact of the work. With the suggested revisions, the paper could become a valuable resource for advancing long-context training methods for LLMs.

---

### Decision · Program_Chairs · 2025-01-22

Accept (Poster)